# Impact of COVID-19 on 1-Year Survival Outcomes in Hepatocellular Carcinoma: A Multicenter Cohort Study

**DOI:** 10.3390/cancers15133378

**Published:** 2023-06-27

**Authors:** Shuell De Souza, Jeffrey Kahol de Jong, Ylenia Perone, Shishir Shetty, Maria Qurashi, Mathew Vithayathil, Tahir Shah, Paul Ross, Laura Temperley, Vincent S. Yip, Abhirup Banerjee, Dominik Bettinger, Lukas Sturm, Helen L. Reeves, Daniel Geh, James Orr, Benjamin Allen, Robert P. Jones, Rohini Sharma

**Affiliations:** 1Department of Surgery & Cancer, Imperial College London, Hammersmith Hospital, London W12 0NN, UKm.vithayathil@imperial.ac.uk (M.V.); 2National Institute for Health Research Birmingham Liver Biomedical Research Unit and Centre for Liver and Gastrointestinal Research, Institute of Immunology and Immunotherapy, University of Birmingham, Birmingham B15 2TT, UKtahir.shah@uhb.nhs.uk (T.S.); 3Department of Oncology, Guys’ & St. Thomas’ and King’s College Hospitals, London SE1 9RT, UK; 4Barts and the London HPB Centre, Royal London Hospital, Whitechapel E1 1BB, UK; 5Department of Medicine II (Gastroenterology, Hepatology, Endocrinology and Infectious Diseases), Faculty of Medicine, Freiburg University Medical Center, University of Freiburg, 79098 Freiburg, Germany; 6Newcastle University Clinical and Translational Research Institute, Newcastle University, Newcastle upon Tyne NE1 7RU, UK; 7Department of Hepatology, Bristol Royal Infirmary, Bristol BS2 8HW, UK; 8School of Cancer Studies, Institute of Translational Medicine, University of Liverpool, Liverpool L3 5TR, UK

**Keywords:** hepatocellular carcinoma, COVID-19, treatment, survival, pandemic, time-to-treatment

## Abstract

**Simple Summary:**

As a direct consequence of the COVID-19 pandemic, cancer screening programs, management of chronic disease and cancer treatments were either cancelled or delayed. 164 patients with hepatocellular cancer (HCC) were evaluated to understand the impact of the first lockdown on treatment delays, and 1-year survival. We observed that 70% of patients were not treated within 31 days, with the median time to treatment being 49 days. This was particularly true of those with early-stage disease undergoing surgical or interventional radiology treatment. However, we noted that delay to treatment had no impact on mortality at 1-year. Given the diversity of the patients included in the study, we investigated possible factors influencing treatment delay, such as age, ethnicity, and geographical location. None of these factors were found to be associated with treatment delays. Future studies with longer follow-up are needed to assess whether delay to treatment had a negative impact on patients’ survival. Our study allows a better understanding of treatment delays and short-term impact on survival during severe health services disruptions. Findings from this study highlight the need for better prioritisation of HCC treatment services particularly for the management of early-stage disease during any future pandemic lockdown.

**Abstract:**

Introduction: The COVID-19 pandemic has caused severe disruption of healthcare services worldwide and interrupted patients’ access to essential services. During the first lockdown, many healthcare services were shut to all but emergencies. In this study, we aimed to determine the immediate and long-term indirect impact of COVID-19 health services utilisation on hepatocellular cancer (HCC) outcomes. Methods: A prospective cohort study was conducted from 1 March 2020 until 30 June 2020, correlating to the first wave of the COVID-19 pandemic. Patients were enrolled from tertiary hospitals in the UK and Germany with dedicated HCC management services. All patients with current or past HCC who were discussed at a multidisciplinary meeting (MDM) were identified. Any delay to treatment (DTT) and the effect on survival at one year were reported. Results: The median time to receipt of therapy following MDM discussion was 49 days. Patients with Barcelona Clinic Liver Cancer (BCLC) stages-A/B disease were more likely to experience DTT. Significant delays across all treatments for HCC were observed, but delay was most marked for those undergoing curative therapies. Even though severe delays were observed in curative HCC treatments, this did not translate into reduced survival in patients. Conclusion: Interruption of routine healthcare services because of the COVID-19 pandemic caused severe delays in HCC treatment. However, DTT did not translate to reduced survival. Longer follow is important given the delay in therapy in those receiving curative therapy.

## 1. Introduction

The global emergence of SARS-CoV-2 resulted in the implementation of unprecedented Government lockdowns or declared states of emergency to prevent the spread of disease and limit morbidity and mortality [1]. The COVID-19 pandemic placed a significant burden on healthcare systems, with resources being diverted to the management of patients with COVID-19. Consequently, cancer screening programs, management of chronic disease and cancer treatments were either cancelled or delayed. It is anticipated that COVID-19–related hospital measures may lead to an extra 33,890 cancer-related deaths in the United States alone [2].

HCC is a major cause of cancer-related deaths worldwide, and unlike other cancer types, incidence and mortality rates continue to rise [3]. Treatment and prognosis are stage-dependent and involve transplantation, resection, radiofrequency ablation, chemo- or radioembolisation, or systemic therapy [4]. Therefore, any delays in receipt of therapy may adversely affect clinical outcomes [5]. Though preliminary data suggest minimal impact of COVID-19 on the reduction of overall survival (OS) in patients with HCC [6], the impact on outcomes indirectly through health services disruption is unknown. All HCC surveillance programs ceased during the COVID-19 lockdown period. Moreover, given the significant backlog of radiology requests and lack of inpatient beds, surgery and locoregional therapies were disrupted. Medical treatments, except oral systemic therapy, were discontinued in some centres, with international consensus statements issuing guidance that systemic therapy should be administered to early-stage disease [7], for which there is a distinct lack of evidence. Surgical capacity was significantly reduced as theatre spaces, intensive care facilities, and ventilatory support were reserved to treat critically ill COVID-19 patients and those requiring emergency surgery [1]. Europe experienced its first COVID-19 wave between March 2020 and June 2020, during which there was cessation of cancer services. During the subsequent waves, primary care and hospital care services remained in place piecemeal. SARS-CoV-2 infection has a relatively modest 4.4–5.5% impact on mortality in patients with hepatobiliary cancers [8], with alcoholic liver disease and deranged baseline liver function being independent risk factors for death from hepatocellular cancer (HCC) [9,10]. However, what remains unclear is the impact of healthcare disruptions on clinical outcomes in HCC. We conducted a prospective, multicentre study evaluating the impact of the first wave shut-down on HCC treatment delays defined by NHS treatment targets times and one-year survival outcomes.

## 2. Methods

We conducted a prospective cohort study across eight UK hospitals and one German hospital. All patients with a diagnosis of HCC, confirmed either radiologically or histologically [4], who were discussed at a dedicated hepatobiliary MDM were included in the study. Patients were included regardless of whether they had a new or previous diagnosis of HCC. The majority of MDMs were held online for the duration of the study. The date of patient recruitment corresponded to the commencement of the first pandemic wave, 1 March 2020 until 30 June 2020. Follow-up data were collected until 30 October 2021. Clinical data included demographic data (age, ethnicity, gender), aetiology of liver disease, cirrhosis status, and Child–Pugh score. HCC-related clinical data included date of diagnosis, number and size of HCC nodules, presence of metastases and portal vein thrombosis, BCLC disease stage, and any treatments received. Any delays to treatment (DTT), duration of delay, and any outcomes of treatment delays were recorded. DTT were calculated from the date of MDM decision to treat to the actual receipt of treatment. DTT was defined based on NHS cancer targets, whereby patients should receive first-line definitive treatment within 31 days from diagnosis [11]. Prior to the pandemic, these standards were met across all domains, except surgery for the second or subsequent treatment of cancer [12]. Progression-free survival (PFS) and overall survival (OS) were calculated from the date of diagnosis to the date of progression, following MDM-specified treatment or date of death. Where the date of death was not available, the date of the last follow-up was taken. All methods were carried out in accordance with the declaration of Helsinki.

### Statistical Analysis

SPSS Statistics for Macintosh Version 28.0 (IBM Corp, Armonk, NY, USA) and GraphPad Prism V.9.3 (GraphPad Software, La Jolla, CA, USA) was used to carry out statistical analysis. Kolmogorov–Smirnov test for normality was conducted on the dataset to test for normal distribution. Differences between categorical datasets were assessed using the cross tabs feature using Pearson’s χ^2^ test. For differences between continuous data sets, unpaired *t*-tests and Mann–Whitney U-tests were used for parametric and non-parametric data, respectively. Survival analysis was conducted by Kaplan–Meier method using log rank. Cox regression was also performed to assess the significance of potential factors influencing overall survival.

## 3. Results

During the first wave, 173 patients were discussed at dedicated hepatobiliary MDMs. Nine patients did not have the date of therapy and, therefore, were excluded from the analysis. Of the 164 patients, 64.3% were a new diagnosis of HCC. The majority were white (76.8%) male (84.4%) with cirrhosis (74.3%) secondary to alcohol excess (25.6%), followed by hepatitis C (20.1%). In terms of tumour characteristics, the majority of cases had BCLC-A disease (31.1%) (Table 1).

In terms of treatments received, 20.1% underwent resection, 17.1% radiofrequency ablation, 46.3% transarterial chemo- or radioembolisation (TACE/TARE), 16.5% underwent systemic therapy. Four patients had received stereotactic ablative radiotherapy (SABR) for intermediate-stage HCC (BCLC stage B) and were included in the TACE/TARE group. Of interest, there was a significant difference in the type of treatment received across BCLC stage such that a significant number of patients with early-stage disease received TACE/TARE and systemic therapy (Table 2).

### 3.1. Lockdown Resulted in Significant Delay in Treatment

We investigated delays in any treatment during the first wave. The median time from MDM discussion to commencement of treatment was 49 days (IQR 26–83), with 70.1% of patients commencing treatment after 31 days of MDM discussion. Significant differences in treatment waiting timings were observed across all treatment types (Figure 1). The median number of days between the decision to treat and the start of treatment date in the resection, ablation, TACE/TARE and systemic therapy cohorts were 56, 49, 50.5, and 25 days, respectively. Hence, delays with treatment were more commonly seen with those undergoing surgery and TACE/TARE, with those undergoing systemic therapy less likely to experience a delay (*p* = 0.05). Although it was observed that 36.0% of patients underwent restaging prior to receiving therapy, with a change in BCLC stage observed in 11.6%, the change in staging was not associated with treatment delay (*p* > 0.05).

Given the reported impact of ethnicity on COVID outcomes, we investigated if this is associated with treatment delays. We observed no significant association between ethnicity on DTT. Age was also investigated as a possible predictor of DTT, and again, no association was observed (Table 3). Given the different geographic locations of each treatment centre, the impact of location was assessed. No association between hospitals was observed (*p* = 0.14). Interestingly, small tumour size was associated with DTT on multivariable analysis, HR 0.4 (95%CI: 0.2–1.0, *p* = 0.03). No other association between any other tumour- or liver-related factors correlated with treatment DTT (Table 3).

### 3.2. Delay Had No Impact on Short-Term Mortality

At the time of analysis, 43 patients had progressed. Median PFS was not calculable. DTT had no impact on PFS (Figure 2A), with this lack of difference maintained across different treatments. In terms of survival, ten patients died, again with DTT having no significant impact (Figure 2B).

## 4. Discussion

Whilst the direct impact of SARS-CoV-2 infection on patient outcomes is well documented, what remains unclear is the impact on outcomes as a result of the complete disruption of global healthcare services during the pandemic. All areas of HCC management [13] were impacted, with the suspension of surveillance programs and reduced access to surgical and interventional radiology for cancer treatment, alongside a recommendation that oral tyrosine kinase inhibitors could be considered for all stages of HCC, given that the tablets could be delivered by courier, alleviating the need and risk of attending hospital [14,15]. These changes reflect the rapid reorganisation of hospital activities to minimise patient exposure to SARS-CoV-2 as much as possible, whilst maintaining clinical outcomes [16]. Adoption of guidance regarding kinase inhibitors was confirmed by our study, with these prescribed across all BCLC stages, with no delays seen in the delivery of systemic therapy. As per international guidance [17], in some UK centres, surgical and loco-regional therapies continued for selected patients managed within an MDM setting, assessing the balance of risks versus potential benefit.

We observed that a significant number of patients were not treated within the recommended 31-day wait time, with a median time to treatment of 1.6 months. This was particularly true of those undergoing resection, ablation, and TACE/TARE, reflecting a shift in the allocation of radiology, surgical, and intensive care personnel resources during the pandemic. Geh and colleagues audited HCC services in the Northeast and Cumbria during the pandemic year compared to the previous year and reported a median treatment time from MDM discussion of 1.6 months, consistent with our findings [18]. They also reported larger tumours at presentation and a lower number of new presentations attributed to disruption of HCC surveillance. Similarly, a large Japanese study reported a reduced number of new patient referrals during the pandemic year compared to previous years [19].

Of key significance during the first wave of the pandemic was the delay in receipt of curative therapy in those with early-stage disease. Waiting times for treatment have been shown to have a direct impact on cancer survival outcomes [20,21,22], with every four-week delay in surgery having a direct bearing on cancer mortality [5]. NHS cancer waiting times for hepatobiliary surgery were significantly delayed during the pandemic [12]. The number of patients undergoing surgery within the 31-day target fell from 91% in 2019 to 88% in 2020 across all tumour types [12]. Consistent with NHS England data, we observed a significant delay in surgery. This may be attributable to significant shortages in theatre staff due to redeployment to intensive care units and excess staff sickness absences [23,24]. Some centres recommended that locoregional therapies, such as RFA, be instituted in preference to surgery to reduce the need for postoperative stay in intensive care and hospitalisation time [10]. However, our data suggest delays across all interventional treatments, reflecting guidance that all non-urgent and elective procedures be postponed [25,26]. Furthermore, a recent systemic review, including 62 studies investigating the impact of COVID-19 delays on cancer care, illustrated significant delays in diagnosis and treatment during the first wave [27]. We considered whether DTT impacted on stage migration within the first three months of the pandemic, and whilst a change in staging occurred in a small number of patients, this was not attributable to DTT in our relatively small study.

Given the diversity of our study cohort, we investigated possible factors influencing treatment delay, in particular ethnicity. Significant disparities in HCC surveillance and treatment are recognised between the White population and minorities ethnic groups, with the Black population in the USA having consistently higher mortality rates from HCC [28]. Whilst similar data are not available in Europe, it was clear that Black, Asian, and Minority Ethnic individuals had higher mortality rates from COVID-19 [29]. Moreover, primary healthcare services are less accessible to those in ethnic minority communities [30], a situation made worse by the closure of general practice during the pandemic, which resulted in fewer presentations of patients with symptoms during the duration of the first wave. Our cohort was reflective of a UK population [31]. We did not observe any differences in DTT between ethnicities. This may be attributable to our short follow-up time, and future work should consider late presentation and tumour stage across ethnic backgrounds stemming from the pandemic. In addition, we found no relationship between DTT and age. The elderly were advised to shield during the pandemic, which may have further exacerbated late presentation [1], and it would be of interest to investigate rates of late-stage presentations across age in future work.

A study conducted by Lai and colleagues reported that 7165 excess cancer deaths occurred within one year of diagnosis in England, which was attributed to treatment delays to COVID-19 [2], but DTT did not affect overall survival. One explanation for these findings and ours is the short follow-up time (one year).

Relapse following curative therapies for HCC occurs in 60–70% within five years [4], therefore, one year will not accurately reflect any changes in clinical outcome as a result of DTT. Thus, a longer follow-up (minimum two years) similar to the median survival of patients with BCLC stage B disease might yield statistically significant results.

There are a number of limitations to our work. Patients were only recruited from major HCC treatment centres. The impact of the pandemic on smaller, regional centres would be of key interest as disruptions to services may have had a greater impact on these hospitals as most services in major teaching remained open albeit attenuated [11]. Recruitment was inconsistent from each site, and with only four patients recruited from Liverpool Hospital NHS Foundation Trust, there may be significant selection bias present. Similarly, only one centre from Germany was included in the study, which might not be reflective of the HCC population of other regions within Germany. Furthermore, our follow-up time was only one year, longer follow-up would be of interest to assess whether DTT had a negative impact on treatment outcomes. Nonetheless, we have conducted a prospective, multicentre study that allows a better understanding of DTT for HCC treatment during severe health services disruptions. Findings from this study highlight the need for better prioritisation of HCC treatment services during times of complete lockdown in the future. Moreover, considering that significant delays were observed in curative HCC treatments, data from this study could provide essential information on the significance of modifying HCC services to maximise survival outcomes during a pandemic. Although the impact of COVID-19 on HCC surveillance is outside the scope of this study, several studies have emphasised the importance of minimum disruption to diagnostic and surveillance services during periods of healthcare crisis. This will ensure that the populations’ needs are met and prevent backlog, fuelling increased diagnosis of HCC at advanced symptomatic stages.

## 5. Conclusions

The COVID pandemic resulted in significant DTT across all HCC treatments but was of particular concern in those with early-stage disease. The delays observed are attributable to the rapid and prompt reorganisation of services to optimise patient care for those most unwell whilst minimising the risk of exposure to COVID-19 for others. Whilst the adoption of telemedicine has been universal, future strategies to avoid DTT may include modifying surgical and interventional radiology workflows so that cancer hubs are created to reduce exposure to infection whilst maintaining treatment targets.

## Figures and Tables

**Figure 1 cancers-15-03378-f001:**
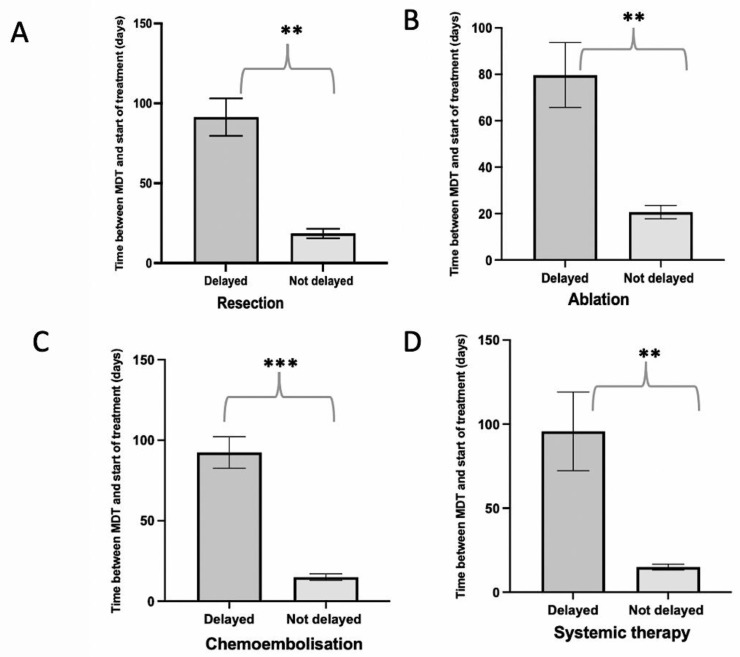
Effect of the 3-month COVID-19 complete lockdown on treatment waiting times across all HCC treatments. Bar graphs illustrate the effect on COVID-19 lockdown on time between decision to treat (DTT) MDM date and start date of treatment across treatments. In panels (**A**–**D**), the bars represent the mean time between MDM DTT and start of treatment, and the error bars represent the standard error of the mean (SEM). Statistically significant data comparisons with *p* < 0.01 are denoted with ** and *p* < 0.001 are denoted with ***. Delay in treatment was defined according to the 31 days NHS target.

**Figure 2 cancers-15-03378-f002:**
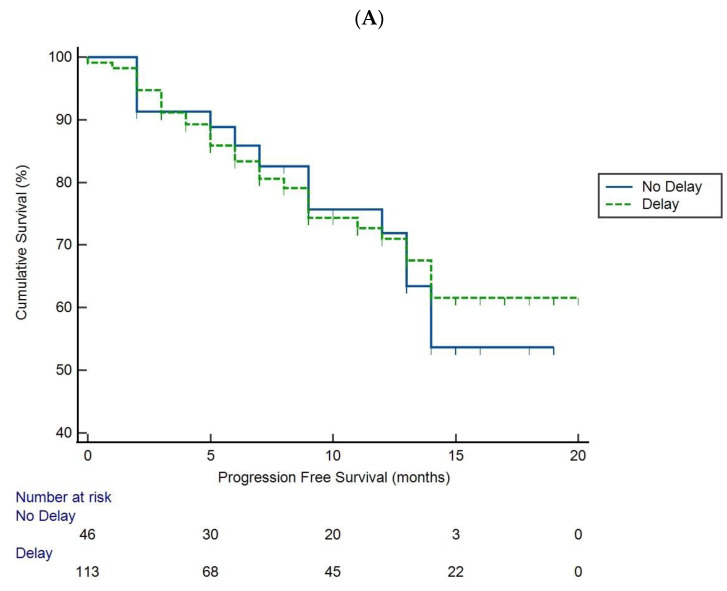
Kaplan–Meier curves illustrating the impact of delayed treatment on (**A**) progression-free survival (PFS) and (**B**) overall survival (OS) in HCC with patients diagnosed during COVID-19 complete lockdown period.

**Table 1 cancers-15-03378-t001:** Clinical and demographic characteristics of patients included in the study.

		Yes (115)	No (49)	*p*-Value
Sex	Male	99 (86.1%)	41 (83.7%)	*p* = 0.71
	Female	16 (13.9%)	8 (16.3%)	
Age	≤70 years	64 (55.7%)	29 (59.2%)	*p* = 0.68
	>70 years	51 (44.3%)	20 (40.8%)	
Ethnicity	White	91 (84.3%)	35 (76.1%)	*p* = 0.229
	Non-white	17 (15.7%)	11 (23.9%)	
Aetiology	HBV	8 (7.0%)	3 (6.1%)	*p* = 0.52
	HCV	25 (21.7%)	8 (16.3%)	
	NASH/NAFLD	27 (23.5)	12 (24.5)	
	ARLD	28 (24.3%)	14 (28.6%)	
	Other	13 (11.3%)	10 (20.4%)	
Tumour diameter in mm (median)		41.33 ± 2.86 (30.5)	50.57 ± 5.69 (35.5)	*p* = 0.11
BCLC score	0-A	40 (34.7%)	11 (22.5%)	*p* = 0.008 *
	B	24 (20.9%)	9 (18.4%)	
	C	8 (7.0%)	9 (18.4%)	
	D	3 (2.6%)	3 (6.1%)	
	Unknown	40 (34.8)	17 (34.7%)	
Metastatic disease	Yes	6 (5.2%)	8 (16.3%)	*p* = 0.020 *
	No	109 (94.8%)	41 (83.7%)	
ECOG Performance score	0–1	70 (60.8%)	31 (63.2%)	*p* = 0.47
	2	2 (1.7%)	2 (4.1%)	
	3	2 (1.7%)	0 (0%)	
	Unknown	41 (35.7%)	16 (32.7%)	
Child– Pugh class	A	89 (77.4%)	40 (81.6%)	*p* = 0.72
	B	15 (13.0%)	5 (10.2%)	
	C	1 (0.9%)	1 (2.0%)	
	Unknown	10 (8.7%)	3 (6.1%)	

Categorical data are presented as frequency with percentage in parenthesis. Pearson’s χ^2^ test for significance was performed for categorical data sets. Continuous data are presented as mean ± standard deviation and median in the parenthesis. Mann–Whitney U test or unpaired *t*-test was performed to assess for significance for continuous data sets. Statistically significant data comparisons with *p*-values of <0.05 are denoted with *. HBV, hepatitis B virus; HCV, hepatitis C virus; NASH, non-alcoholic steatohepatitis; NAFLD, non-alcoholic fatty liver disease; ARLD, alcohol-related liver disease; CLD, chronic liver disease; BCLC, Barcelona Clinic Liver Criteria; ECOG, Eastern Co-operative Oncology Group classification.

**Table 2 cancers-15-03378-t002:** Treatments received by BCLC stage.

		Surgery	Ablation	TACE/TARE	Systemic Therapy
BCLC score	A	16 (69.6%)	14 (87.5%)	20 (40.0%)	1 (5.6%)
	B	4 (17.4%)	2 (12.5%)	22 (44.0%)	5 (27.8%)
	C	1 (4.3%)	0 (0.0%)	6 (12.0%)	10 (55.6%)
	D	2 (8.7%)	0 (0.0%)	2 (4.0%)	2 (11.1%)
	Total	23 (100.0%)	16 (100.0%)	18 (100.0%)	50 (100.0%)

**Table 3 cancers-15-03378-t003:** Factors impacting on delay to treatment on univariable and multivariable analysis.

	Univariable Model		Multivariable Model	
	Hazard Ratio (95% CI)	*p*-Value	Hazard Ratio (95% CI)	*p*-Value
Delay (yes vs. no)	1.1 (0.6–2.1)	0.7		
Ethnicity (White vs. not White)	1.4 (0.7–2.7)	0.3		
Aetiology (Viral vs. Non-viral)	0.7 (0.4–1.4)	0.4		
BCLC Stage (A/B vs. C/D)	0.4 (0.2–0.8)	0.01	0.5 (0.2–1.2)	0.2
CTP Class (B/C vs. A)	1.1 (0.5–2.7)	0.8		
Tumour size <7 cm	0.4 (0.2–0.7)	0.002	0.4 (0.2–1.0)	0.03
Metastatic disease	0.5 (0.2–1.2)	0.1		

## Data Availability

The datasets generated and/or analysed during the current study are not publicly available due to the clinical nature of the data but are available from the corresponding author on reasonable request.

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
