# Peer review of "Impact of COVID-19 on 1-Year Survival Outcomes in Hepatocellular Carcinoma: A Multicenter Cohort Study"

_cancers, 2023, doi:10.3390/cancers15133378_

Round 1
Reviewer 1 Report (Previous Reviewer 1)
It is my third time to review the manuscript. I still cannot see great improvement on the statistical analysis. I suggest authors carefully solving below concerns before submitting the revised version.
1. On line 103, authors mentioned that 9 were excluded from the 171 patients. However, the final sample size was 164 instead of 162. Why ?
2. Is table 3 the univariable or multivariable analysis ? I suggest presenting two tables for the two analysis.
3. On line 158, authors mentioned that the impact of DTT was explored in a multiple variable analysis with other known predictive factors of PFS. I did not see related table in the manuscript.
4. Why is table 1 the same as table 3 ?
Minor editing of English language required.
Author Response
Thank you and apologies that there are outstanding issues remaining with our manuscript. We have addressed these in full below and as track changes within the manuscript. We would be happy to make any further changes as required.
It is my third time to review the manuscript. I still cannot see great improvement on the statistical analysis. I suggest authors carefully solving below concerns before submitting the revised version.
- On line 103, authors mentioned that 9 were excluded from the 171 patients. However, the final sample size was 164 instead of 162. Why ?
We apologise for this error 173 patients were identified, nine were excluded leaving 164. We have corrected the error on line 103.
- Is table 3 the univariable or multivariable analysis ? I suggest presenting two tables for the two analysis.
Table 3 has been replaced with the correct table that illustrates both univariable and multivariable analysis. We have used one table for ease of reading.
- On line 158, authors mentioned that the impact of DTT was explored in a multiple variable analysis with other known predictive factors of PFS. I did not see related table in the manuscript.
This line has been removed. DTT was not predictive of PFS and therefore its independence compared with other known factors of prognosis were not included.
- Why is table 1 the same as table 3 ?
Table 3 is incorrect and this should be the univariable and multivariable data. This has now been corrected.
Reviewer 2 Report (Previous Reviewer 2)
Dear Authors
I would like to thank you for the opportunity of reviewing again this interesting paper that is focused on a very remarkable and challenging topic that is a lively argument also in daily clinical practice.
The COVID-19 pandemic placed a significant burden on healthcare systems, with resources being diverted to the management of patients with COVID-19. As a consequence, cancer screening programs, management of chronic disease and cancer treatments were either canceled or delayed. However, the impact of healthcare disruptions on clinical outcomes in HCC still remains unclear. The present prospective, multicentre study evaluated the impact of the first wave shut-down on HCC treatment delays defined by NHS treatment targets times and 1-year survival outcomes.
The Authors addressed almost all the raised points appropriately. However, they did not cite the suggested reference when they discussed the optimization and implementation of both intra and post-procedural workflows [Hepatoma Res 2022;8:27. doi:10.20517/2394-5079.2022.18]. Finally, please follow the Journal guidelines for reference in the text (reference numbers should be placed in square brackets [] and placed before the punctuation) and in the Reference section (Author 1, A.B.; Author 2, C.D. Title of the article. Abbreviated Journal Name Year, Volume, page range)
Best regards,
Author Response
Thank you and apologies that there are outstanding issues remaining with our manuscript. We have addressed these in full below and as track changes within the manuscript. We would be happy to make any further changes as required.
It is my third time to review the manuscript. I still cannot see great improvement on the statistical analysis. I suggest authors carefully solving below concerns before submitting the revised version.
- On line 103, authors mentioned that 9 were excluded from the 171 patients. However, the final sample size was 164 instead of 162. Why ?
We apologise for this error 173 patients were identified, nine were excluded leaving 164. We have corrected the error on line 103.
- Is table 3 the univariable or multivariable analysis ? I suggest presenting two tables for the two analysis.
Table 3 has been replaced with the correct table that illustrates both univariable and multivariable analysis. We have used one table for ease of reading.
- On line 158, authors mentioned that the impact of DTT was explored in a multiple variable analysis with other known predictive factors of PFS. I did not see related table in the manuscript.
This line has been removed. DTT was not predictive of PFS and therefore its independence compared with other known factors of prognosis were not included.
- Why is table 1 the same as table 3 ?
Table 3 is incorrect and this should be the univariable and multivariable data. This has now been corrected.
I would like to thank you for the opportunity of reviewing again this interesting paper that is focused on a very remarkable and challenging topic that is a lively argument also in daily clinical practice.
The COVID-19 pandemic placed a significant burden on healthcare systems, with resources being diverted to the management of patients with COVID-19. As a consequence, cancer screening programs, management of chronic disease and cancer treatments were either canceled or delayed. However, the impact of healthcare disruptions on clinical outcomes in HCC still remains unclear. The present prospective, multicentre study evaluated the impact of the first wave shut-down on HCC treatment delays defined by NHS treatment targets times and 1-year survival outcomes.
The Authors addressed almost all the raised points appropriately. However, they did not cite the suggested reference when they discussed the optimization and implementation of both intra and post-procedural workflows [Hepatoma Res 2022;8:27. doi:10.20517/2394-5079.2022.18].
Included
Finally, please follow the Journal guidelines for reference in the text (reference numbers should be placed in square brackets [] and placed before the punctuation) and in the Reference section (Author 1, A.B.; Author 2, C.D. Title of the article. Abbreviated Journal Name Year, Volume, page range)
We have used the recommended style by the journal. We do hope this is correct.

Round 2
Reviewer 1 Report (Previous Reviewer 1)
Thank you for your response. I have no other comments.
Minor editing of English language required.
This manuscript is a resubmission of an earlier submission. The following is a list of the peer review reports and author responses from that submission.
Round 1
Reviewer 1 Report
The authors investigated the DTT during the pandemic and its impact on clinical outcomes in hepatocellular carcinoma. The major conclusions are not innovative and solid given the small sample size and short follow-up period. Statistical analyses can also be improved.
Major comments:
1. Since the patients were enrolled in 2020, why were the long-term OS and PFS data not collected and analyzed?
2. Please give detailed results for the log-rank test in Figure 2, including events, censoring rate, and estimated 1-year PFS or OS rate for each group.
3. It is recommended to conduct a multivariate logistic regression to test the predictors in Table 1.
4. I didn’t see the Cox model result in the paper.
5. The authors treated delay as a dichotomous variable in the paper. How about the impact of DTT time (continuous variable) on OS and PFS in the delayed patients?
Minor Comments:
1. Please provide test statistics and delayed treatment rates for every categorical level in Table 1.
2. Please combine the plots for easy comparison in Figure 1.
Author Response
The authors investigated the DTT during the pandemic and its impact on clinical outcomes in hepatocellular carcinoma. The major conclusions are not innovative and solid given the small sample size and short follow-up period. Statistical analyses can also be improved.
Thank you for reviewing our manuscript. Whilst there has been a number of papers considering the impact of Covid on outcomes in HCC, these are generally limited to single hospital institutions. We have reported a UK wide experience of the impact of covid on outcomes in patients with HCC. The follow-up period is short which we acknowledge. Future work will report on the long-term outcomes.
Major comments:
- Since the patients were enrolled in 2020, why were the long-term OS and PFS data not collected and analyzed
We report the 1year outcome data. Despite our best efforts, it was not possible to collate all the survival data for the remaining patients from the centers in a timely manner. As suggested by reviewer 2, we have changed the title of the paper to reflect the 1 year follow-up
- Please give detailed results for the log-rank test in Figure 2, including events, censoring rate, and estimated 1-year PFS or OS rate for each group.
This has been addressed and Figure 2 A and B updated accordingly
- It is recommended to conduct a multivariate logistic regression to test the predictors in Table 1.
We have included a cox regression model (Table 3) and this is now discussed in the results.
- I didn’t see the Cox model result in the paper.
This has been added (table 3)
- The authors treated delay as a dichotomous variable in the paper. How about the impact of DTT time (continuous variable) on OS and PFS in the delayed patients?
We did consider this however this was not included as there is no international standard definition of “DTT”. As outlined, NHS treatment target is 31 days and this was taken forward in this study.
Minor Comments:
- Please provide test statistics and delayed treatment rates for every categorical level in Table 1.
The statistics used in the study are defined in the statistical section of the paper. Table 1 is a demographics table and is purely descriptive. No statistical tests were used.
- Please combine the plots for easy comparison in Figure 1
We feel that combining all the plots will clutter the figure making it difficult to read.
Reviewer 2 Report
Dear Authors
I would like to thank you for the opportunity of reviewing this interesting paper that is focused on a very remarkable and challenging topic that is a lively argument also in the daily clinical practice.
The COVID-19 pandemic placed a significant burden on healthcare systems, with resources being diverted to the management of patients with COVID-19. As a consequence, cancer screening programs, management of chronic disease and cancer treatments were either cancelled or delayed. However, the impact of healthcare disruptions on clinical outcomes in HCC still remains unclear. The present prospective, multicentre study evaluated the impact of the first wave shut-down on HCC treatment delays defined by NHS treatment targets times, and 1-year survival outcomes.
Papers that explore in depth this theme, especially in the era of COVID-19 pandemic, could surely be of interest for this important journal. Moreover, this paper demonstrates the aim of finding objective and practical conclusions from the many studies that have been conducted in recent years.
This paper is pleasurable to read, although it suffers from some limitations that Authors can easily adjust in order to slightly improve their review making it more eligible for this important Journal. Furthermore, Authors can improve some section of the paper, adding information and including other important references about this topic that, in my opinion, should be cited and discussed.
First of all, I believe the title could be improved and the fact that only 1-year survival outcomes were assessed should be specified: “Impact of COVID-19 on 1-year survival outcomes in Hepatocellular carcinoma: A multicentre cohort study”.
Moreover, please correct the affiliation for Dr. Helen L Reeves.
I believe Authors did not correctly reported keywords from MeSH Browser (for example, “Cancer care” is not present. This is important, in my personal opinion, in order to increase the traceability of this paper (and consequently the possibility of the Journal to be cited by Readers and Stakeholders).
Lines 65-70, please improve the scientific soundness of this part.
Please check reference 12.I don’t think it is necessary and, maybe, even not appropriate.
In the “Methods” section, regarding the dedicated hepatobiliary (MDM) and the patients included in the study. Which are the characteristics of patients discussed in the MDM? They were complicated cases or every HCC cases? They were all patients with an already known HCC diagnosis or also with newly diagnosed HCC? They were all already in systemic therapy and/or with previous surgical/loco-regional treatments? Please explain.
Moreover, given the increasingly widespread possibility to revise diagnostic images and conduct multidisciplinary meetings online [JHEP Rep. 2020 Jun;2(3):100113. doi: 10.1016/j.jhepr.2020.100113] [Hepatology. 2020 Jul;72(1):287-304. doi: 10.1002/hep.31281], how were the MDMs held? online and/or in person?
Lines 90-91, “DTT was defined based on NHS cancer targets, whereby patients should receipt first-line definitive treatment within 31 days from diagnosis.” The definition is correct. However, it would be interesting to compare the data obtained in this study with those in the pre-COVID era. Were the standards required by NHS maintained before the pandemic? If yes, the Authors should specify it.
Table 2. Please add statistical analysis in the Table.
In the “Discussion”, please discuss more deeply the evidence that the delays with treatment were more commonly seen with those undergoing surgery and TACE/TARE, with those undergoing systemic therapy; in particular, could the Authors please try to explain why systemic therapy has experienced fewer delays than other treatment choices? In fact, despite the indications for surgery, locoregional therapy, and liver transplantation in HCC patients have not changed following the COVID-19 outbreak, a significant difference in the modification of the treatment strategy was noted. Probably, these results are due to the rapid and prompt reorganization of activities following the pandemic outbreak that was made to minimize its effect on patient outcomes and reduce the risk of exposure to SARS-CoV-2 as much as possible. These changes have probably primarily affected both loco-regional and surgical therapies due to the lack of beds and personnel [Int J Mol Sci. 2023 Jan 6;24(2):1091. doi: 10.3390/ijms24021091]. Please cite the aforementioned reference and discuss this topic.
In conclusion, the pandemic has led to a rapid and prompt reorganization of activities in order to minimize its effect on patient outcomes and reduce the risk of exposure to SARS-CoV-2 as much as possible. Since the optimization and implementation of both intra and postprocedural workflows seems to be the right path to follow, interventional radiologists and surgeons should adapt themselves to these changes as well [Hepatoma Res 2022;8:27. doi:10.20517/2394-5079.2022.18]. Please, briefly discuss what should be done in the future to overcome the delay in therapies due to pandemic.
Please add in the limits sections that the follow-up was performed only for 1-year.
Finally, please follow the Journal guidelines for reference in the text (reference numbers should be placed in square brackets [] and placed before the punctuation) and in the Reference section (Author 1, A.B.; Author 2, C.D. Title of the article. Abbreviated Journal Name Year, Volume, page range)
Best regards,
Author Response
Dear Authors
I would like to thank you for the opportunity of reviewing this interesting paper that is focused on a very remarkable and challenging topic that is a lively argument also in the daily clinical practice.
The COVID-19 pandemic placed a significant burden on healthcare systems, with resources being diverted to the management of patients with COVID-19. As a consequence, cancer screening programs, management of chronic disease and cancer treatments were either cancelled or delayed. However, the impact of healthcare disruptions on clinical outcomes in HCC still remains unclear. The present prospective, multicentre study evaluated the impact of the first wave shut-down on HCC treatment delays defined by NHS treatment targets times, and 1-year survival outcomes.
Papers that explore in depth this theme, especially in the era of COVID-19 pandemic, could surely be of interest for this important journal. Moreover, this paper demonstrates the aim of finding objective and practical conclusions from the many studies that have been conducted in recent years.
This paper is pleasurable to read, although it suffers from some limitations that Authors can easily adjust in order to slightly improve their review making it more eligible for this important Journal. Furthermore, Authors can improve some section of the paper, adding information and including other important references about this topic that, in my opinion, should be cited and discussed.
Thank you for your kind comments
First of all, I believe the title could be improved and the fact that only 1-year survival outcomes were assessed should be specified: “Impact of COVID-19 on 1-year survival outcomes in Hepatocellular carcinoma: A multicentre cohort study”.
We have taken the suggestion onboard.
Moreover, please correct the affiliation for Dr. Helen L Reeves.
Corrected
I believe Authors did not correctly reported keywords from MeSH Browser (for example, “Cancer care” is not present. This is important, in my personal opinion, in order to increase the traceability of this paper (and consequently the possibility of the Journal to be cited by Readers and Stakeholders).
Thank you for this helpful suggestion. These have been altered
Lines 65-70, please improve the scientific soundness of this part.
This has been altered
Please check reference 12.I don’t think it is necessary and, maybe, even not appropriate.
This was inserted in error and has been removed
This has been corrected
In the “Methods” section, regarding the dedicated hepatobiliary (MDM) and the patients included in the study. Which are the characteristics of patients discussed in the MDM? They were complicated cases or every HCC cases? They were all patients with an already known HCC diagnosis or also with newly diagnosed HCC? They were all already in systemic therapy and/or with previous surgical/loco-regional treatments? Please explain.
We have added a line to say that both new and previous diagnosis of HCC were included. We mention in the results that 64% had a new diagnosis of HCC. All clinical data were collected.
Moreover, given the increasingly widespread possibility to revise diagnostic images and conduct multidisciplinary meetings online [JHEP Rep. 2020 Jun;2(3):100113. doi: 10.1016/j.jhepr.2020.100113] [Hepatology. 2020 Jul;72(1):287-304. doi: 10.1002/hep.31281], how were the MDMs held? online and/or in person?
We have added a line to say that the MDMs were held online and in person
Lines 90-91, “DTT was defined based on NHS cancer targets, whereby patients should receipt first-line definitive treatment within 31 days from diagnosis.” The definition is correct. However, it would be interesting to compare the data obtained in this study with those in the pre-COVID era. Were the standards required by NHS maintained before the pandemic? If yes, the Authors should specify it.
This has been added “Prior to the pandemic, these standards were met across all domains, bar surgery for second or subsequent treatment of cancer”
Table 2. Please add statistical analysis in the Table.
This is a descriptive table illustrating treatments received across BCLC stages. No formal statistical analysis was undertaken.
In the “Discussion”, please discuss more deeply the evidence that the delays with treatment were more commonly seen with those undergoing surgery and TACE/TARE, with those undergoing systemic therapy; in particular, could the Authors please try to explain why systemic therapy has experienced fewer delays than other treatment choices? In fact, despite the indications for surgery, locoregional therapy, and liver transplantation in HCC patients have not changed following the COVID-19 outbreak, a significant difference in the modification of the treatment strategy was noted. Probably, these results are due to the rapid and prompt reorganization of activities following the pandemic outbreak that was made to minimize its effect on patient outcomes and reduce the risk of exposure to SARS-CoV-2 as much as possible. These changes have probably primarily affected both loco-regional and surgical therapies due to the lack of beds and personnel [Int J Mol Sci. 2023 Jan 6;24(2):1091. doi: 10.3390/ijms24021091]. Please cite the aforementioned reference and discuss this topic.
This suggestion has been incorporated into the discussion and references added as suggested
In conclusion, the pandemic has led to a rapid and prompt reorganization of activities in order to minimize its effect on patient outcomes and reduce the risk of exposure to SARS-CoV-2 as much as possible. Since the optimization and implementation of both intra and postprocedural workflows seems to be the right path to follow, interventional radiologists and surgeons should adapt themselves to these changes as well [Hepatoma Res 2022;8:27. doi:10.20517/2394-5079.2022.18]. Please, briefly discuss what should be done in the future to overcome the delay in therapies due to pandemic.
This suggestion has been incorporated into the discussion and references added as suggested
Please add in the limits sections that the follow-up was performed only for 1-year.
This has been added. “Furthermore, our follow-up time was only 1-year – longer follow-up would be of interest to assess whether DTT had a negative impact on treatment outcomes.”
Finally, please follow the Journal guidelines for reference in the text (reference numbers should be placed in square brackets [] and placed before the punctuation) and in the Reference section (Author 1, A.B.; Author 2, C.D. Title of the article. Abbreviated Journal Name Year, Volume, page range)
This has been corrected
Round 2
Reviewer 1 Report
The authors clarified part of the issues. However, I am still confused about the tables and figures presented.
1. Please give detailed results for the log-rank test in Figure 2, including events, censoring rate, and estimated 1-year PFS or OS rate for each group.
This has been addressed and Figure 2 A and B updated accordingly
New comment: I still did not see the events, censoring rate, and estimated 1-year PFS or OS rate for each group, P value either in Figure 2 or in the text. It is not enough to give a K-M plot without other statistics.
2. It is recommended to conduct a multivariate logistic regression to test the predictors in Table 1
We have included a cox regression model (Table 3) and this is now discussed in the results.
New comment: Outcomes in Table 1 is dichotomous. You should conduct a multivariate logistic regression instead of Cox model. In addition, why is Table 1 the same as Table 3 ?
3. I didn’t see the Cox model result in the paper.
This has been added (table 3)
New comment: Table 3 is not the Cox model result. You mentioned “Cox regression was also performed to assess the significance of potential factors influencing overall survival.” I did not see the result this time.
Reviewer 2 Report
Authors addressed raised points appropriately.